# A Review of Margetuximab-Based Therapies in Patients with HER2-Positive Metastatic Breast Cancer

**DOI:** 10.3390/cancers15010038

**Published:** 2022-12-21

**Authors:** Moudi M. Alasmari

**Affiliations:** 1College of Medicine, King Saud Bin Abdul Aziz University for Health Sciences (KSAU-HS), Jeddah 21461, Saudi Arabia; asmarim@ksau-hs.edu.sa; 2King Abdullah International Medical Research Centre (KAIMRC), Jeddah 21423, Saudi Arabia

**Keywords:** margetuximab, Fc engineering, clinical trials, anti-HER2 therapy, human epidermal receptor 2, positive metastatic breast cancer

## Abstract

**Simple Summary:**

HER2+ metastatic breast cancer (MBC) is a highly prevalent type of breast cancer owing to its resistance to conventional anti-HER2 drugs. Therefore, novel agents that can arrest tumor progression and enhance the overall survival rates of HER2+ breast cancer patients, which would represent a major advancement toward the treatment of HER2 + MBC, need to be developed. This review provides insights into the clinical implication and utility of margetuximab, an anti-HER2 drug, in HER2 + MBC treatment, focusing on studies on the efficacy of margetuximab. Margetuximab is indeed an excellent addition to the arsenal of anti-HER2 mAb drugs that can be used for treating HER2 + MBC.

**Abstract:**

Breast cancer (BC) is the most commonly diagnosed cancer globally, with high mortality rates. Targeted drug therapies have revolutionized cancer treatment. For example, treatment with human epidermal receptor 2 (HER2) antagonists has markedly improved the prognosis of patients with HER2-positive BC (HER2 + BC). However, HER2+ metastatic BC (MBC) remains prevalent owing to its resistance to conventional anti-HER2 drugs. Therefore, novel agents are needed to overcome the limitations of existing cancer treatments and to enhance the progression-free and overall survival rates. Progress has been made by optimizing the fragment crystallizable (Fc) domain of trastuzumab, an IgG1 monoclonal, chimeric anti-HER2 antibody, to develop margetuximab. The modified Fc domain of margetuximab enhances its binding affinity to CD16A and decreases its binding affinity to CD32B, thereby promoting its antitumor activity. This review summarizes studies on the efficacy of margetuximab, discusses its utility as an anti-HER2 monoclonal antibody drug for the treatment of HER2 + BC, and presents the latest advances in the treatment of BC. This review provides insights into the clinical implication of margetuximab in HER2 + MBC treatment.

## 1. Introduction

Breast cancer (BC), the most malignant tumor type of cancer in females, accounts for 11.7% of newly diagnosed cancer cases and 6.9% of cancer-related deaths annually [1]. Tumor heterogeneity is regarded as one of the hallmarks of cancers, including BC [2]. Owing to its molecular heterogeneity, BC is differentiated into four subtypes based on the presence or absence of the following biomarkers: hormone receptors (HRs), human epidermal receptor 2 (HER2), and Ki-67 (cell proliferation marker). The four subtypes of BC are: (I) luminal A (HR+/HER2−/low Ki-67), (II) luminal B (HR+/HER2−/high Ki-67) or (HR+/HER2+/any Ki-67), (III) non-luminal HER2+ (HR-/HER2 overexpression), and (IV) triple-negative (HR−/HER2−) along with the basal-like subtype [3,4]. These subtypes vary in etiology, risk factor profile, organ site preference for metastases, responses to therapy, and prognoses [5].

HER2 (also known as ErbB2, C-erbB2, or Her2/neu) is a tyrosine kinase protein belonging to the epidermal growth factor receptor (EGFR) family [6]. HER2 expression levels, depicting HER2 positivity (+) or negativity (−), are used as diagnostic and prognostic biomarkers for solid tumors [7]. HER2 positivity is confirmed based on the score of 3+ in ≥10% of tumor cells in the immunohistochemistry (IHC) test or a HER2/chromosome enumeration probe (CEP17) ratio of ≥2 in the in-situ hybridization (ISH) assay along with an average HER2 copy of <4 signals per cell in the pathological test [6,7]. Compared with other BC subtypes, HER2-positive cancer subtypes are characterized by decreased apoptosis and enhanced cell proliferation, mobility, adherence, invasiveness, angiogenesis, metastasis, and epithelial cell survival [7]. HER2 genomic alterations have been identified in 20–30% of invasive BC cases [8]. Moreover, HER2 + BC is a highly immunogenic and aggressive subtype with poor clinical outcomes and high recurrence rates owing to its resistance to chemotherapy [9]. Conventionally, HER2 + BC tends to grow faster and is associated with an increased risk of the progression of systemic and brain metastases. Tumor characteristics that promote metastasis in BC remain elusive [10]. Approximately 15–20% of BC cases are of the HER2 + BC subtype; the prevalence attributed to HER2 overexpression is a consequence of genomic alteration [11].

When diagnosed during the initial stages, before regional lymph node metastasis, BC is curable. However, after organ metastasis, BC cannot be cured with the current treatment strategies, resulting in high mortality rates. The extent of BC metastases is generally detected by imaging and validated by biopsy [10]. Moreover, tumor age, size, stage, histological type, and lymphovascular status are considered crucial factors for prognostication and treatment considerations [10]. The higher rates of metastasis are the result of the amplification of the ErbB2 gene. HER2 + BC metastasis is most prominently observed in the bones (65–67%), lungs (45–35%), and the brain (30–55%) [12]. It can also metastasize to distant organs such as the auxiliary lymph nodes, liver, and peritoneal cavity [10,12].

Over the last two decades, the progression-free survival (PFS) rates and prognosis of patients with HER2+ metastatic BC (MBC) have substantially improved due to the development of novel anti-HER2 targeted therapies. For example, the monoclonal antibody trastuzumab is the recommended treatment during the initial stages of HER2 + BC; it is either administered alone (monotherapy) or in combination with cytotoxic agents (taxane), followed by doxorubicin therapy [13,14]. Although this treatment shows promising outcomes in terms of survival rates and the reduction in relapse and metastasis, it shows severe cardiac toxicity, especially when used in combination with anthracycline agents, preventing its use in some patients. Moreover, resistance to trastuzumab has been reported in some patients. Consequently, to overcome toxicity, resistance, and other limitations of targeted therapy, novel HER2 inhibitors have been developed [15].

This review summarizes the current knowledge on margetixumab (Margenza^®^), a novel anti-HER2 drug recently approved by the United States Food and Drug Administration (US FDA), with an overview on its mechanism of action and its clinical significance. This review will be helpful for practicing clinicians.

## 2. Structural Biology of HER-Family Receptors

The human epidermal growth factor receptor (EGFR/ErbB) family comprises four members, namely HER1 (ErbB1), HER2 (ErbB2), HER3 (ErbB3), and HER4 (ErbB4), encoded by the *ERBB1, ERBB2, ERBB3*, and *ERBB4*, respectively [16]. These receptors are transmembrane tyrosine kinases with partial homology. Each receptor contains an extracellular domain (ECD) with four subunits (I, II, III, and IV) and a transmembrane lipophilic domain consisting of 19–25 hydrophobic amino acids. The ECD is 630 amino acids long, where the first and third subdomains (I/L1 and III/L2) are leucine-rich segments involved in ligand binding; they exhibit a β-barrel conformation. However, the second and fourth subdomains (II/CR1 and IV/CR2), which facilitate dimer formation, are cysteine-rich domains with disulfide bonds. Domain IV harbors a cleavage site for matrix metalloproteases (MMP) [17].

An Intracellular portion of about 550 amino acid residues containing (i) juxtamembrane segment, (ii) functional tyrosine kinase (TK) domain with catalytic activity, and (iii) cytoplasmic carboxyl-terminal tail with multiple phosphorylation sites, is essential for the activation of downstream signaling pathways [18]. The tyrosine kinase domain is essential for the dimerization of HER receptors. Ligand binding by the receptors is essential for the activation of the tyrosine kinase domains, which facilitates their dimerization. Dimerization can take place between molecules of the same receptor (homodimerization) or between two different HER receptors (heterodimerization) [19,20,21,22].

When inactive, i.e., in the absence of a ligand, these receptors exist as monomers in a tethered conformation due to the intramolecular interactions between domains II and IV. In this tethered conformation the “dimerization arm” (β-hairpin/loop) that exists in cysteine-rich domain II, is completely buried in the tethering arm of domain IV. This restricts the movement of receptor ECD arms and stabilizes the tethered receptor conformation, which is responsible for the auto-inhibition of ligand binding and dimerization [20,23]. However, ligand binding to domains I and III induces a conformational change that destabilizes the intramolecular tether and prevents the autoinhibitory effect. This exposes the dimerization arm and consequently allows dimerization.

Ligand-induced dimerization activates intracellular tyrosine kinase and enables it to assume an asymmetric active structure to create pTyr docking sites via kinase trans-tyrosine phosphorylation. These docking sites lodge the Src homology 2 (SH2) and phosphotyrosine binding (PTB) motifs of phosphotyrosine-binding proteins (e.g., Grb2, Shc, and PLCγ) [23]. The specific interaction of the pTyr-binding motifs with signaling proteins activates downstream signaling pathways, including the Ras-mitogen activated protein kinase (MAPK), Ras/Raf/MEK/MAPK, phosphoinositide-3-kinase (PI3K)-Akt, phospholipase C-gamma (PLC-γ), Src signaling, and signal transducer and activator of transcription (STAT) pathways [20]. These signaling pathways regulate essential cellular processes, including cell proliferation, migration, motility, differentiation, and apoptosis [24]. However, the cellular effects mainly depend on the intermediate pathway that is activated, and the magnitude and duration of ligand binding, which is further diversified by various ligands and the dimer in question [21].

Except for HER2, all ErbB family receptors directly bind to ligands. This is due to HER2’s ECD lacking a known ligand-binding domain [18,19]. It is therefore thought that HER2 functions as a co-receptor with, or dimerization partner of, other ErbB receptors. Similarly, all ErbB receptors exhibit tyrosine kinase activity, except for HER3, which has no or minimal catalytic kinase activity [22]. Interestingly, heterodimerization between HER2 and HER3 exhibits the most potent pro-tumorigenic and mitogenic signaling activity [23].

## 3. Unique Characteristics of HER2 Promote Tumor Progression in BC

The HER2 protein has several fates (Figure 1). For example, as a monomer, as a homodimer, and as a HER2/HER3 or HER2/HER4 heterodimer [19]. The TK domain of HER2 is activated upon homo/hetero dimerization, evoking a signaling cascade that activates the receptive gene, leading to cell proliferation, migration, invasion, or cell survival, which are the hallmarks of cancer. The activation of the signaling cascades contributes to the heterogeneity of different cancer types (e.g., ovarian cancer, BC, and non-small cell lung cancer).

The overexpression/amplification of ErbB receptors shows a significant correlation with poor prognosis, cancer metastasis, decreased survival rates, and enhanced drug resistance [25]. The activation of HER2 signaling pathways is mainly responsible for cellular proliferation and cell survival phenomena, which are regarded as dominant drivers that cause tumor development and progression in nearly 85% of BC cases [26]. The amplification/overexpression of the HER-*neu* proto-oncogene and the HER2/HER3 heterodimer that activates oncoproteins have promiscuous roles in the pathogenesis of solid tumors [27]. Moreover, a truncated HER2 protein that lacks ECD-p95 (p95HER2) and an active C-terminal fragment, is detected in nearly 40% of HER2 + BC cases [28]. Therefore, p95HER2 is used as a predictive biomarker for cancer prognosis. Moreover, it is used to evaluate the efficacy of, or resistance to, existing treatments for BC [29,30].

## 4. Existing Anti-HER2 Therapies for HER2 + BC

Therapies for HER2 + BC primarily target the well-investigated HER1 and HER2 receptors and their pathways. BC cases exhibiting the overexpression of HER2 are clinically aggressive, showing moderate responses to chemotherapy. However, the development of personalized anti-HER2 treatment strategies has revolutionized the treatment of HER2 + BC, showing promising survival rates and improved patient outcomes [31,32,33,34,35]. Therapies include monoclonal antibodies (mAbs; e.g., trastuzumab and pertuzumab), antibody-drug conjugates (e.g., ado-trastuzumab emtansine, trastuzumab-deruxtecan), and tyrosine kinase inhibitors (e.g., lapatinib and neratinib) (Table 1) [13,32,33,34,35,36,37,38,39,40,41,42,43,44,45,46].

## 5. Current Insights on Margetuximab, a Novel Anti-HER2 Drug for Treatment of Positive Metastatic Breast Cancer

Research to develop anti-HER2-personalised therapeutic agents is being conducted at a rapid pace, which is evident based on the number of molecules being introduced into clinical trials. Rituximab initiated an era of immunotherapeutics, with mAbs extensively employed afterward to target tumors [47,48,49,50]. The therapeutic functionality of mAbs depends on the interactions of two regions thereof with components of the host immune system: the fragment antigen binding (Fab) region that binds to the antigen and the fragment crystallizable (Fc) region that interacts with the FcγRs and C1 complex (C1q) components of the immune system [51,52,53,54]. The mAbs exert their cytotoxic actions by promoting the interaction between Fc and FcγRs to activate the innate immune response by engaging complement-dependent cytotoxicity (CDC), antibody-dependent cellular phagocytosis (ADCP), and antibody-dependent cellular cytotoxicity (ADCC) [51,55,56,57].

Antibody-dependent cellular cytotoxicity involves a cascade of mechanisms that target the FcRIIIa (CD16A) receptor on the cell surface and the Fc domains of immunoglobulins. It acts by enhancing natural killer (NK) cells, monocytes or macrophages (CD16+ subsets), and NKT cells or γδ T cells to enhance cytolysis and exert antitumor effects [58,59,60,61,62,63]. The FcγR family in humans comprises activating receptors and inhibitory receptors. Activating receptors include FcγRI (CD64, a high-affinity receptor), FcγRIIa (CD32a), FcγRIIIa (CD16a), and GPI-linked FcγRIIIb (low affinity); inhibitory receptors include FcγRIIb. The CD16A, CD32A, and CD32B receptors are expressed on effector cells and regulate immune activation processes. Human CD16a is a transmembrane low-affinity IgG Fc receptor [64] that triggers immune NK cells for their ADCC effects via the lone FcγR present in the NK cells [65,66]. Mutations in the FcγRIIIa gene generate two FcγRIIIa polymorphs with valine (V) and phenylalanine (F) at amino acid position 158 (FcγRIIIA-V158F). This polymorphism is considered crucial as it influences the rate of tumor cell lysis via ADCC, with the high-affinity valine *v/v* allele being responsible for more lysis than the V/F or FF alleles [51,52].

HER2 + MBC was successfully treated using the humanized IgG1 antibody, trastuzumab [53]. This was the first anti-HER2 mAb approved by the US FDA. It hinders HER1 activity by modulating extracellular HER2-*neu* [54]. Specifically, trastuzumab binds to the ECD of the HER2 receptor to inhibit its homodimerization and destabilize its heterodimers. Furthermore, it inhibits the formation of p95HER2, implicated in tumor progression. Moreover, trastuzumab can mediate ADCC via the activation of NK cells, made possible by the detection of the Fc portion thereof, ultimately leading to the death of cancer cells. Trastuzumab triggers HER2 internalization followed by lysosomal degradation and activates the c-Cbl-ubiquitin ligase-mediated ubiquitination and degradation of HER2. Trastuzumab also prevents matrix metalloproteinase (MMP)-mediated HER2 shedding. All of these effects ultimately lead to the inhibition of downstream signaling cascades [55,56]. The pathways inhibited by trastuzumab include the PI3K pathway, where decreasing phosphatase and tensin Homologue (PTEN) phosphorylation and AKT dephosphorylation increases cell death. Trastuzumab also inhibits the MAPK pathway, which activates the cyclin-dependent kinase inhibitor p27 KIP1 and promotes cell-cycle arrest and apoptosis. These actions show that trastuzumab exhibits anti-proliferative and anti-angiogenic effects, ultimately leading to the death of cancer cells. However, a major setback of this drug is the occurrence of drug resistance [54].

Novel immunotherapeutic agents are being developed to overcome resistance and enhance the overall survival of HER2 + BC patients. One such drug, margetuximab (MGAH22), was authorized by the US FDA on 16 December 2020 [57,58].

## 6. Pharmacology of Margetuximab

Fc-engineering strategies have been used over the years to customize mAbs to enhance their cytotoxic and antitumor potencies, margetixumab is the consequence of that effort. This drug, developed by MacroGenics, is a novel IgG1 monoclonal human/mouse chimeric antibody engineered in its Fc-domain to target HER2 (Figure 2) and is derived from 4D5, a precursor to trastuzumab [59]. The Fc-engineered domain of margetuximab exhibits mutations of five amino acid components (L235V, F243L, R292P, Y300L, and P396L) [67,68,69].

### 6.1. Mechanism of Action

Similar to Trastuzumab, Margetixumab works by binding to the Fab epitope of the HER2 receptor with comparable specificity and affinity (Figure 3) [68] and exhibits Fc-independent antiproliferative effects. It shows elevated relative affinity towards both allelic variants of CD16A. The low-affinity allelic variant (FcγRIIIA-V158F) was found to be associated with a decreased clinical response to trastuzumab. The Fc-engineered domain of margetuximab is specifically optimized to increase its binding to all FcγRIIIA-V158F allelic variants, compared to the wild-type IgG1 [60,70]. This increased binding to FcγRIIIA is associated with the enhanced ADCC activity of human NK cells, which leads to the suppression of cell proliferation [71]. Additionally, the Fc-engineered domain of margetuximab lowers its affinity or decreases its binding to the inhibitory receptor CD32B (FcγRIIB) [53,72]. These altered binding capabilities of margetuximab, particularly in cells with lower levels of HER2, in cells resistant to trastuzumab, and in patients bearing FcγRIIIA-V158F, result in increased ADCC and enhanced anti-tumor effects [73].

### 6.2. Pharmacokinetic Properties

The molecular formula of margetuximab is C_6484_H_10010_N_1726_O_2024_S_42_ [74]. Its pharmacokinetics have been thoroughly investigated in Phase 1 clinical trials using pharmacokinetic two-compartmental models with parallel linear and Michaelis–Menten elimination. The approved dose used for the pharmacokinetic studies was 15 mg/kg Q3W or 6.0 mg/kg QW [73]. Bang et al. (2017) reported that the pharmacokinetic parameters of clearance (CL), central volume (V1), inter-compartment clearance (Q), and peripheral volume (V2) were 0.299, 3.73, 0.885, and 3.73 L/day, respectively. Moreover, the authors reported the distribution (t_.5_ dist) and half-life (t_.5_) at 1.12 and 15.5 days, respectively [73]. The C_max_ was 466 µg/mL (20%), whereas the AUC0–21d was 4,120 µg/day/mL (21%) following administration to patients with advanced HER2 + BC and relapsed or refractory status, with a volume of distribution of 5.47 L (22%). Margetuximab is metabolized through several catabolic pathways into smaller peptides via proteases in a pattern similar to that of the other mAbs. The terminal half-life of Margetuximab was 19.2 days (28%), with a clearance rate of 0.22 L/day (24%). An approximately 3% concentration decrease in the serum levels of Margetuximab was observed four months post-discontinuation [61].

### 6.3. Indications/Therapeutic Use

Margetuximab, used synergistically with chemotherapy, is prescribed to adult patients with BC previously treated with two or more anti-HER2 agents, of which at least one was administered to treat metastatic diseases [61].

### 6.4. Tolerability and Toxicity

Treatment with margetuximab is well accepted and tolerated [61,73]. The SOPHIA clinical trial showed the safety profile of margetuximab. In this clinical trial, fatal adverse reactions were reported in 1.1% of margetuximab-receiving patients, with viral and aspiration pneumonia at 0.8% and 0.4%, respectively. Serious adverse reactions occurred in 16% of margetuximab-treated patients. The most prominent of these were left ventricular (LV) dysfunction and infusion-related reactions (IRRs) [61]. Although LV cardiac dysfunction was observed in 1.9% of margetuximab-treated patients, a lack of studies on margetuximab-treated patients with a LV ejection fraction of less than 50% or a history of the cardiac disease led to the addition to the warning and precautions section in FDA drug label [61]. The SOPHIA clinical trial showed that IRRs occurred in 13% of margetuximab-receiving patients. The majority of IRRs were reported during cycle 1 of therapy and resolved within 24 h. These infusion-related reactions were associated with symptoms such as nausea, fever, hypotension, anemia, diarrhea, headache, vomiting, fatigue, tachycardia, and certain cutaneous manifestations such as urticaria or rash. However, therapy for 9% of margetuximab-receiving patients was interrupted due to IRRS, and IRRS was discontinued in margetuximab therapy in 0.4% of treated patients. Patients experiencing severe infusion-related reactions must be carefully screened. Several toxicity parameters, including risk assessment during pregnancy and lactation, were not evaluated. Data on margetuximab-associated toxicity remains limited [61].

### 6.5. Dosage and Administration

A dosage of 15 mg/kg has been approved for administration via intravenous infusion. The initial dose is administered over 120 min, followed by another for approximately 30 min every three weeks until disease progression or unacceptable toxicity is observed. For therapy using a combined regimen, Margetuximab should be administered immediately following the completion of chemotherapy [61,75].

### 6.6. Clinical Trials of Margetuximab in BC

Clinical trials of Margetuximab began in 2010 and were conducted by MacroGenics [60,73,76]. A phase 1 trial (NCT01148849) was developed to assess the safety, efficacy, pharmacokinetic properties, pharmacodynamics, and antitumor activities of Margetuximab [73]. A total of 66 patients were included, all of whom presented with HER2-overexpressing breast and/or gastric carcinomas. Patients were divided into two groups, whereby 34 patients were treated with regimen A (intravenous infusion at a dose range of 0.1–6.0 mg/kg weekly for 3 weeks), while 32 were treated with regimen B (intravenous infusion at a dose range of 10–18 mg/kg once every 3 weeks). Margetuximab was well tolerated at the given doses without any attainable maximum tolerated dose for either regimen. The trial results indicated that, of the 60 response-evaluable patients, 12% showed partial responses, while 52% reached stable disease. Moreover, a reduction in tumor size was observed in 78% of patients. Adverse effects were mild, with constitutional symptoms such as nausea and pyrexia. The authors concluded that Margetuximab exhibited promising activity and could be used alone to treat patients with HER2+ tumors. This trial facilitated further research on the potential clinical application of Margetuximab as a single agent or in combination therapy [73,77].

Nordstrom et al. (2011) reported that margetuximab exhibited a promising safety profile, with 0.1 mg/kg as a minimum human equivalent dose in trials on cynomolgus monkeys receiving 150 mg/kg [60]. ‘No observed adverse effect level’ (NOAEL) was observed; the dose was derived from the minimum effective dose of 1 mg/kg used in transgenic mice bearing human CD16A-V158F (xenograft models). Furthermore, margetuximab treatment was considered acceptable at a dose range of 15–150 mg/kg [60]. This study was followed by a phase II clinical trial (NCT01828021) that employed a Simon 2-stage design in 41 patients to investigate the efficacy and activity of margetuximab in patients with advanced BC, in either relapsed or refractory status, and low HER2 expression, as evidenced using a fluorescence ISH test [78]. A series of trials involving Chinese patients has also been conducted (NCT04398108 and NCT04262804). The most prominent clinical trial was a phase III trial (SOPHIA, NCT02492711). This randomized, parallel assignment, open-label trial design comprised 536 patients aged ≥ 18 years with confirmed HER2 + MBC or unresectable BC previously treated with at least two HER2-directed therapies in the metastatic setting [76]. A 1:1 randomization was performed based on the chemotherapy treatment (capecitabine, gemcitabine, eribulin, or vinorelbine) administered along with margetuximab (15 mg/kg) or trastuzumab (6 mg/kg; loading dose, 8 mg/kg) over three-week cycles. The authors reported that PFS was enhanced in patients receiving margetuximab compared to those receiving trastuzumab, with a median of 5.8 vs. 4.9 months (95% CI, 4.2–5.6; hazard ratio [HR], 0.76; *p* = 0.033; 95% CI, 0.59–0.98 at 24% relative risk reduction). The median overall survival (OS) was 21.6 months with margetuximab vs. 19.8 months with trastuzumab (HR, 0.89; 95% CI, 0.69–1.13; *p* = 0.33). The final results of the SOPHIA trial reported superior overall survival (OS) in HER2 + MBC with CD16A-158F low-affinity allele patients in comparison to the trastuzumab-treated group where the median OS was 23.6 vs. 19.2 months, respectively (HR 0.72; 95% CI 0.52-1.00; nominal *p* = 0.05). Although this trial exhibited a comparable safety profile between margetuximab and trastuzumab-treated groups, IRRs were commonly seen in the margetuximab-treated group (13.6%) compared to the trastuzumab group [79]. Margetuximab was approved following a series of clinical trials (Table 2) to treat HER2 + MBC in December 2020. Margetuximab is currently undergoing several other clinical trials to further investigate its usage, safety, and efficacy in other cancer types, such as gastric and gastroesophageal junction cancer [80,81]. The trials currently registered with Clinicaltrials.gov are listed in Table 3.

## 7. Conclusions

ErbB2 receptors are capable of enhancing the malignancy of solid tumors, including breast and gastric cancers, and sustaining cancer types resistant to conventional therapies. The scientific community deemed HER2 an effective target for cancer treatment, prevention, and diagnosis. The development of novel cancer-targeted therapies is a major advancement toward the treatment of HER2 + MBC. A step towards this was achieved by optimizing the Fc domain of trastuzumab to create the novel mAb margetuximab. Clinical trials with margetuximab have demonstrated its efficacy in treating HER2 + MBC. Importantly, compared with the standard trastuzumab combined with chemotherapy, the combination of margetuximab with chemotherapy showed favorable results in terms of overall response rate (ORR) and PFS. However, further evidence is required to determine its optimal use in a variety of clinical settings. Studies on treatment resistance to this drug and toxicity profiling are also warranted.

## Figures and Tables

**Figure 1 cancers-15-00038-f001:**
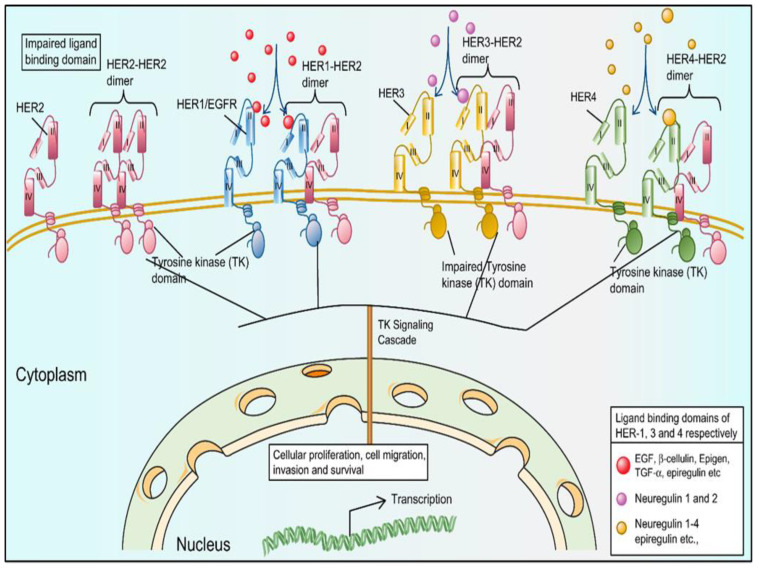
The HER family of tyrosine kinases (TK) receptors have various ligand-binding capabilities that help orchestrate several biological processes in normal cells. The HER family of receptors comprises HER1, HER2, HER3, and HER4, depicted in blue, magenta, golden, and green, respectively. All members of the HER receptor family are structurally similar, containing an extracellular domain with a ligand-binding site (except HER2), a lipophilic transmembrane domain, and a TK intracellular domain (except for HER3). The TK domain of HER1, HER2, and HER4 is activated following homo- or heterodimerization and evokes a signaling cascade that activates a receptive gene, leading to cell proliferation, migration, invasion, or cell survival.

**Figure 2 cancers-15-00038-f002:**
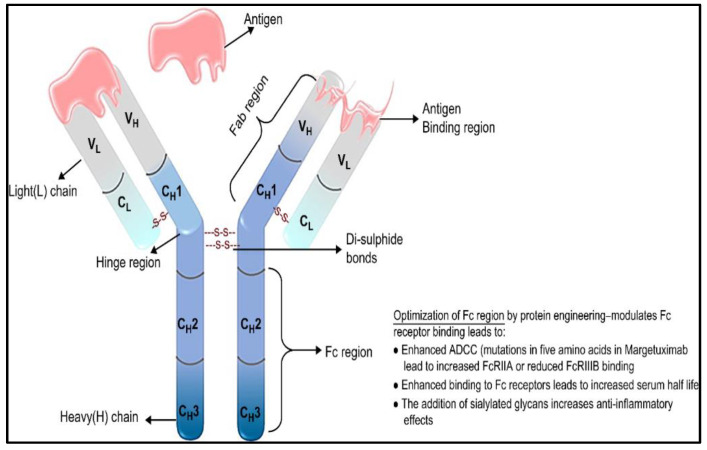
Schematic representation of margetuximab. Margetuximab is a chimeric IgG1 kappa mAb with an optimized Fc region to enhance binding. The Fab and the Fc regions of IgG1 are indicated using brackets. Heavy chain domains are represented by C_H_1 to C_H_3, whereas light chain domains are denoted by C_L_. V_H_ and V_L_ represent the variable heavy chain and variable light chain, respectively.

**Figure 3 cancers-15-00038-f003:**
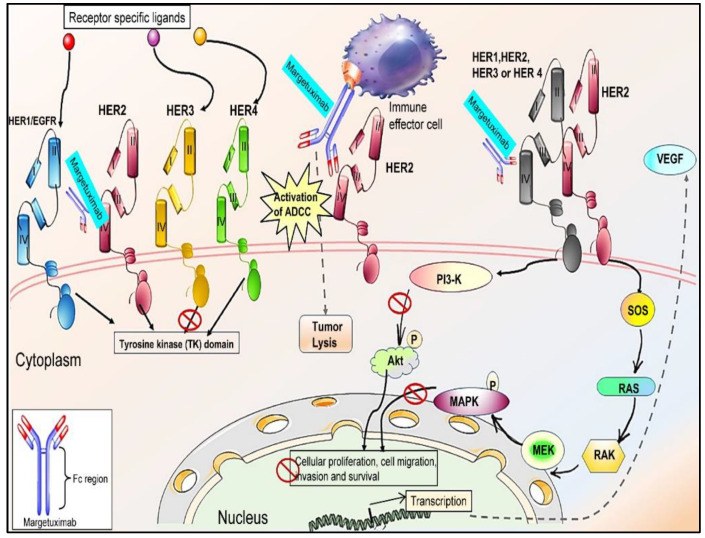
Schematic representation of the mechanism underlying the action of margetuximab. Margetuximab is an Fc-optimized mAb that binds to ECD IV of the HER2 receptor, thus preventing the formation of HER2–HER2 homodimers and ligand-independent HER2–HER3, HER2–HER1, and HER2–HER4 complexes/heterodimers. It enhances ADCC, leads to tumor lysis, and inhibits the TK signaling cascade, thus limiting cancer cell proliferation, migration, invasion, and survival. Figures were generated using Microsoft PowerPoint and free templates obtained from https://smart.servier.com/ (accessed on 9 June 2022).

**Table 1 cancers-15-00038-t001:** Anti-HER2 targeted therapies approved by the US FDA.

**Agent**	**Year of FDA** **Approval**	**Mechanism** **of Action**	**Indication**	**Dosage**	**Major** **Adverse** **Effects**	**Box Warning**
Trastuzumab [31,32]	1998	A humanized monoclonal IgG1 antibody that targets the extracellular domain (domain IV) of human HER2/neu, preventing dimerization and inducing ADCC	Treatment of HER2/neu overexpressing breast cancer/HER2 + MBC (first-line setting in therapy); used as adjuvant/neo adjuvant therapy alongside chemotherapy for ≤ 1 year.	Initial loading dose of 4 mg/kg IV followed by 2 mg/kg weekly or a loading dose of 8 mg/kg IV followed by 6 mg/kg every 3 weeks	Fever, infusion related reactions, diarrhea, headache, increased cough, rash, anemia, neutropenia, and myalgia	Cardiotoxicity, decline in left ventricular function; pulmonary toxicity (rare); infusion-related reactions
Lapatinib [33,34,35]	2007	Reversible tyrosine kinase inhibitor of HER1 and HER2 phosphorylation, resulting in an inhibition of signal transduction	Used in combination with capecitabine	1250 mg/kg orally for 1–21 days along with 1000 mg/m^2^ capecitabine for 1–14 days, repeated every 3 weeks	Nausea, diarrhea, fatigue, and rash (acne, dermatitis acneiform)	Idiosyncratic hepatotoxicity
Pertuzumab [36,37]	2012	A humanized monoclonal IgG1 antibody that targets the ECD II of HER2/neu, preventing heterodimerization of HER2 with other HER family members	Used as adjuvant/neo adjuvant therapy; given for 1 year for node positive disease; used in conjunction with trastuzumab and taxane for MBC as a first-line therapy	Initial loading dose of 840 mg/kg followed by 420 mg/kg every 3 weeks	Nausea, diarrhea, neutropenia, alopecia, rash, and peripheral neuropathy	Cardiotoxicity, decline in LVF, IRR hypersensitivity reactions/anaphylaxis, cardiomyopathy, and embryo-fetal toxicity
Ado-trastuzumab Emtansine (T-DM1) [38,39]	2013	An antibody drug conjugate comprising trastuzumab linked to a potent anti-microtubule agent DM1 (derivative of maytansine); causes cell-cycle arrest, leads to apoptosis, induces ADCC, and disrupts downstream HER2 signaling	Used to treat patients with HER2 + MBC who previously received trastuzumab and a taxane; adjuvant treatment of patients with HER2+ early BC	3.6 mg/kg IV every 3 weeks	Fatigue, nausea, headache, thrombocytopenia, and constipation	Hepatotoxicity, left ventricular dysfunction, pulmonary toxicity, IRR hypersensitivity reactions, thrombocytopenia, and neurotoxicity
Afatinib [40,41]	2013	Irreversible HER1 and HER2 tyrosine kinase inhibitor; induces phosphorylation leading to subsequent inhibition of signal transduction	Used as first-line therapy for advanced NSCLC patients with mutant-HER1	Afatinib, 40 mg or 30 mg once daily	Diarrhea, paronychia, acneiform skin rash, stomatitis, and a loss of appetite	Left ventricular dysfunction, diarrhea, hepatotoxicity, hand-foot skin reaction, and interstitial lung disease
Neratinib [42,43,44,45]	2017	Irreversible HER1 and HER2 tyrosine kinase inhibitor; causes phosphorylation leading to subsequent inhibition of signal transduction	Extended adjuvant therapy administered after trastuzumab and chemotherapy	240 mg orally with food continued for 1 year along with prophylactic loperamide	Diarrhea, nausea, vomiting, anorexia, abdominal pain fatigue, and decreased appetite	Diarrhea (grade ≥ 3), hepatotoxicity, and embryo-fetal toxicity
Trastuzumab-deruxtecan [46]	2019	An antibody-drug conjugate comprising trastuzumab linked to topoisomerase I inhibitor (deruxtecan); induces ADCC, disrupts downstream HER2 signaling leads to apoptosis and cell-cycle arrest	Used for patients with metastatic or unresectable HER2+ breast cancer who have had one or more prior anti-HER2-based regimens	5.4 mg/kg IV every 3 weeks	Nausea, diarrhea, vomiting, musculoskeletal pain, and myelosuppression	Interstitial lung disease, left ventricular dysfunction, neutropenia, embryo-fetal toxicity

Abbreviations: ADCC, antibody-dependent cellular cytotoxicity; BC, breast cancer; EGFR, epidermal growth factor; HER2, human epidermal growth factor receptor 2; IRR, infusion-related reactions; LVF, left ventricular dysfunction; NSCLC, non-small cell lung cancer; MBC, metastatic breast cancer.

**Table 2 cancers-15-00038-t002:** Clinical trials involving Margetuximab in breast cancer patients.

NCT Identifier	Year of Clinical Study	Study Title	Phase and Study Design	Study Participant	Study Type	Subject Number	Status	Study Arm
NCT01148849 [73]	2010	Safety study of Margetuximab in HER2+ carcinomas	**I**-Single Group Assignment, open label, treatment purpose	≥18 years (adults, older adults), with confirmed HER2 + MBC	IV	66	Completed	Margetuximab
NCT01828021 [60]	2013	Phase 2 study of Margetuximab in patients with relapsed or refractory advanced BC	**II**-Single Group Assignment, open label, treatment purpose	Age ≥ 18 years (adults, older adults), with confirmed invasive BC	IV	25	Completed	Margetuximab
NCT02492711 [76]	2015	Margetuximab plus chemotherapy vs. Trastuzumab plus chemotherapy in the treatment of HER2 + MBC (SOPHIA)	**III**-Randomized, parallel assignment, open label, treatment purpose	Age ≥ 18 years (adults, older adults), with confirmed HER2 + MBC	IV	624	Completed	Margetuximab and the chosen chemotherapy (Capecitabine/Vinorelbine/Eribulin/Gemcitabine) vs. Trastuzumab and the chosen chemotherapy
NCT03133988	2017	Margetuximab Expanded Access Program	not available	Children, adults, older adults	EA	Case -by-case basis	Approved for marketing	Margetuximab
NCT04262804	2020	A study to evaluate the efficacy and safety of Margetuximab plus chemotherapy in the treatment of Chinese patients with HER2 + MBC	**II**-Randomized, parallel assignment, open label, treatment purpose	Male or female, age ≥ 18 years, with confirmed HER2 + MBC; have received at least 2 prior lines of anti-HER2 directed therapy in the metastatic setting	IV	120	Recruiting	Margetuximab and the chosen chemotherapy (Capecitabine/Vinorelbine/Gemcitabine) vs. Trastuzumab and the chosen chemotherapy
NCT04398108	2020	A study to evaluate the pharmacokinetics of Margetuximab in Chinese patients with HER2 + MBC	**I**-Single group assignment, open label, treatment purpose	Male or female, age ≥ 18 years, with confirmed HER2 + MBC; have received at least 2 prior lines of anti-HER2 directed therapy in the metastatic setting	IV	16	Completed	Margetuximab and the chosen chemotherapy (Capecitabine/Vinorelbine/Gemcitabine)
NCT04425018	2020	MARGetuximab or trastuzumab (MARGOT) (MARGOT)	**II**-Randomized, parallel assignment, open label, treatment purpose	Male or female, age ≥ 18 years, with confirmed Stage II or III invasive BC	IV	171	Recruiting	Arm (a): Paclitaxel, Pertzumab, and Margetuximab; arm (b): Paclitaxel, Pertzumab, and Trastuzumab

Note: ClinicalTrials.gov entries as of September 2021 are listed. Abbreviations: IV, Interventional; EA, Expanded Access; BC, breast cancer; MBC, metastatic breast cancer.

**Table 3 cancers-15-00038-t003:** Clinical trials involving margetuximab in other types of cancer patients.

NCT Identifier	Study Title	Phase and Study Design	Study Participant	Study Type	Subjects, n	Status	Study Arm	Indications
NCT02689284 [80]	Combination of margetuximab and pembrolizumab for advanced, metastatic HER2+ gastric or gastroesophageal junction cancer	**Ib/2** Single group assignment, open label, treatment purpose	Age ≥ 18 years’ old; with confirmed HER2 + MGEJ or gastric cancer; Have received trastuzumab or at least 1/>lines of cytotoxic CT in the metastatic setting	Interventional	95	Completed	Margetuximab plus pembrolizumab	Gastric, stomach, and esophageal cancer
NCT04082364 [81]	Combination of margetuximab, INCMGA00012, MGD013, and chemotherapy Phase 2/3 trial in HER2+ gastric/GEJ cancer (MAHOGANY)	**II/III** Randomized, parallel assignment, open label, treatment purpose	Age ≥ 18 years (Adult, Older Adult), with confirmed HER2 + M (GEJ) or gastric cancer	Interventional	860	Recruiting	Cohort A: single-arm cohort. (safety efficacy evaluation); Cohort B Part 1: randomized, 4-arm segment; Cohort B Part 2: randomized, 2-arm segment	Gastric cancer, gastroesophageal junction cancer, and HER2+ gastric cancer
NCT03219268 [76]	A study of MGD013 in patients with unresectable or metastatic neoplasms	**I** Non-randomized, single group assignment, open label, treatment purpose	Age ≥18 years (Adult, Older Adult), with confirmed advanced unresectable or metastatic solid tumors; cohort expansion only	Interventional	353	Active, not recruiting	Dose escalation followed by Cohort Expansion Phase at the MTD. MGD013; MGD013 + margetuximab	Advanced solid tumors, hematologic neoplasms, ovarian cancer, HER2 + BC, NSCLC, cervical cancer, TNBC etc.

Note: ClinicalTrials.gov entries as of September 2021 are listed. Abbreviations: BC, breast cancer; CT, chemotherapy; NSCLC, non-small cell lung cancer; HER2 + MGEJ, HER2+ metastatic gastroesophageal junctional cancer; TNBC, triple-negative breast cancer.

## Data Availability

All data are contained within the article.

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
