# Peer review of "A Review of Margetuximab-Based Therapies in Patients with HER2-Positive Metastatic Breast Cancer"

_cancers, 2022, doi:10.3390/cancers15010038_

Round 1

Reviewer 1 Report

The review highlights the cancer-targeted therapies using monoclonal antibodies for the treatment of HER2-positive metastatic breast cancer, especially monoclonal antibody margetuximab. Most of the review highlights HER2+ breast cancer, HER2 receptors, and current treatment options with very few details regarding the studies using margetuximab. The authors should consider several minor revisions to improve the overall quality of the manuscript.

1. The review is good, however, more details are needed for the outcome of completed clinical studies using margetuximab as a single agent or in combination with other drugs. 

2. The introduction sections 2 and 3 about HER2 and tumor progression can be reduced, as the review should focus more on margetuximab.

3. Include any preliminary studies conducted using margetuximab in in vivo xenograft models, prior to escalating to clinical studies after FDA approval in 2020. 

4. In table 2, include the year of the clinical study. 

Author Response

  1. The review is good, however, more details are needed for the outcome of completed clinical studies using margetuximab as a single agent or in combination with other drugs. 

Response: Thank you for your comment. As per your suggestion, I have added the following in section 6.6:

The most prominent clinical trial was a phase III trial (SOPHIA, NCT02492711). This randomized, parallel assignment, open-label trial comprised 536 patients aged ≥ 18 years with confirmed HER2+ MBC or unresectable BC previously treated with at least two HER2-directed therapies in the metastatic setting [76]. A 1:1 randomization was performed based on the chemotherapy treatment (capecitabine, gemcitabine, eribulin, or vinorelbine) administered along with margetuximab (15 mg/kg) or trastuzumab (6 mg/kg; loading dose, 8 mg/kg) over three-week cycles. The authors reported that PFS was enhanced in patients receiving margetuximab compared to those receiving trastuzumab, with a median of 5.8 vs. 4.9 months (95 % CI, 4.2–5.6; hazard ratio [HR], 0.76; p = 0.033; 95 % CI, 0.59–0.98 at 24 % relative risk reduction). The median overall survival (OS) was 21.6 months with margetuximab vs. 19.8 months with trastuzumab (HR, 0.89; 95 % CI, 0.69–1.13; p = 0.33). The final results of the SOPHIA trial reported superior overall survival (OS) in HER2+ MBC with CD16A-158F low-affinity allele patients in comparison to the trastuzumab-treated group where the median OS was 23.6 vs 19.2 months, respectively (HR 0.72; 95% CI 0.52-1.00; nominal P=0.05). Although this trial exhibited a comparable safety profile between margetuximab and trastuzumab-treated groups, IRRs were commonly seen in the margetuximab-treated group (13.6%) compared to the trastuzumab group [79].  

  1. The introduction sections 2 and 3 about HER2 and tumor progression can be reduced, as the review should focus more on margetuximab.

Response: I appreciate your comment but the other reviewer has asked to add more information to these parts, so I am trying to not extend it more. 

  1. Include any preliminary studies conducted using margetuximab in in vivoxenograft models, prior to escalating to clinical studies after FDA approval in 2020. 

Response:  Nordstrom et al. (2011) reported that margetuximab exhibited a promising safety profile, with 0.1 mg/kg as a minimum human equivalent dose in trials as cynomolgus monkeys receiving 150 mg/kg [60]. ‘No observed adverse effect level’ (NOAEL) was observed; the dose was derived from the minimum effective dose of 1 mg/kg used in transgenic mice bearing human CD16A-V158F (xenograft models). Furthermore, margetuximab treatment was considered acceptable at a dose range of 15–150 mg/kg [60]. This study was followed by a phase II clinical trial (NCT01828021) that employed a Simon 2-stage design in 41 patients to investigate the efficacy and activity of margetuximab in patients with advanced BC, in either relapsed or refractory status, and low HER2 expression, as evidenced using a fluorescence ISH test [78].

  1. In table 2, include the year of the clinical study. 

Response: I added the year of clinical study column.

Reviewer 2 Report

The review submitted by Moudi M Alasmari entitled "A Review of Margetuximab-Based Therapies in Patients with HER2-Positive Metastatic Breast Cancer" aims to give an overview about the benefits of using an optimized fragment crystallizable (Fc) domain of trastuzumab, an IgG1 monoclonal, chimeric anti-HER2 antibody, called margetuximab. This review summarizes the studies published about the efficacy of margetuximab, discusses its utility as an anti-HER2 monoclonal antibody drug for the treatment of HER2+breast cancer, and sum up the latest advances in the treatment of BC (e.g. resistance mechanisms).

The manuscript is well-written, in a clear form which facilitates the reading. However, in the opinion of this reviewer, the author, in the introduction section, should add some information about the current state of knowledge about the HER-2 mammary carcinoma in dogs and cats, since they can be used as models, given to the genetic proximity. Indeed, FMC is bring a high number of profits to human medicine that should be added to the manuscript (e.g. 10.3390/cancers12061386;
10.1016/j.bbcan.2021.188587
; 10.1186/1746-6148-6-5; 10.18632/oncotarget.7551; 10.1007/s11033-022-07383-4).

Author Response

Response: Thank you for your comment. In this manuscript, I have attempted to emphasize more on the aim of reviewing margetuximab as a new therapy for patients with HER2-positive metastatic breast cancer and I mentioned the preclinical studies done on Margetuximab. In this drug transgenic mice and Monkeys have been used for in vivo studies and I wrote about it in the revised manuscript in section 6.6 lines (336-353) [Nordstrom et al. (2011) reported that margetuximab exhibited a promising safety profile, with 0.1 mg/kg as a minimum human equivalent dose in trials as cynomolgus monkeys receiving 150 mg/kg [60]. ‘No observed adverse effect level’ (NOAEL) was observed; the dose was derived from the minimum effective dose of 1 mg/kg used in transgenic mice bearing human CD16A-V158F (xenograft models). Furthermore, margetuximab treatment was considered acceptable at a dose range of 15–150 mg/kg [60]. This study was followed by a phase II clinical trial (NCT01828021) that employed a Simon 2-stage design in 41 patients to investigate the efficacy and activity of margetuximab in patients with advanced BC, in either relapsed or refractory status, and low HER2 expression, as evidenced using a fluorescence ISH test [78].]

Reviewer 3 Report

Dear author,
thank you for the opportunity to review this manuscript.

The abstract is well written and interesting, and the idea to write a comprehensive review for clinicians to refer to is intriguing. However, unfortunately, I don't think the manuscript can be accepted in its current form.
I suggest to
- Revise mainly the introduction part, where some references are wrongly discussed (e.g. line 51-52 of the paper says "Moreover, the clinical aggression of HER2+ BC is attributed to increased stromal tumor-infiltrating lymphocytes (TILs). [10,11]." However, reference 11 does not investigate this topic, while reference 10 says "...the percentage of TILs is clinically relevant due to its association with better prognosis".), etc.
- Moreover, if the review intended use is for clinicians, it would be good to add how HER2 positivity is depicted (line 40).
- It is important to clearly separate the concept of HER2 positivity and HER2-enriched subtype 
- Moreover, some information were not supported by references: e.g., line 29 (hallmarks of malignant breast carcinoma), etc.
- In paragraph 3, lines 144-148 look redundant to the introduction, and maybe can be more useful there
- Lines 160-161 needs to be checked
- Table 1 is missing Trastuzumab-deruxtecan
- Paragraph 5 only discusses trastuzumab, but ends with a conclusion sentence on margetuximab. Maybe the title and the conclusion need to be changes, or the author needs to discuss the other "monoclonal antidoby-based therapies" too
- The toxicity profile paragraph needs to be clearer: the data are presented in a misleading way and some additional comment from the trials data may be added
- Table 2 needs to be updated, and data from the recently published updated OS of the SOPHIA trial need to be added
- English needs to be revised for mistyping, grammar and syntax mistakes. 

Author Response

- Revise mainly the introduction part, where some references are wrongly discussed (e.g. line 51-52 of the paper says "Moreover, the clinical aggression of HER2+ BC is attributed to increased stromal tumor-infiltrating lymphocytes (TILs). [10,11]." However, reference 11 does not investigate this topic, while reference 10 says "...the percentage of TILs is clinically relevant due to its association with better prognosis".), etc.

Response: The sentence has been omitted to prevent readers from being misled.

- Moreover, if the review intended use is for clinicians, it would be good to add how HER2 positivity is depicted (line 40).

Response: HER2 positivity is confirmed based on the score of 3+ in ≥10% of tumor cells in the immunohistochemistry (IHC) test or a HER2/chromosome enumeration probe (CEP17) ratio of ≥2 in the in situ hybridization (ISH) assay along with an average HER2 copy of <4 signals per cell in the pathological test [6, 7].  

- It is important to clearly separate the concept of HER2 positivity and HER2-enriched subtype

Response: I appreciate your comment. I had already written about the types of breast cancer and added reference 6, which clarifies the difference between them for further reading. However, herein, I think most readers would be familiar with the background of the disease, and the other reviewers had asked to reduce this part.

- Moreover, some information were not supported by references: e.g., line 29 (hallmarks of malignant breast carcinoma), etc.

Response: References have been added accordingly.

- In paragraph 3, lines 144-148 look redundant to the introduction, and maybe can be more useful there

Response: The lines have been removed.

- Lines 160-161 needs to be checked

Response: The lines were redundant; therefore, I have removed it.

- Table 1 is missing Trastuzumab-deruxtecan

Response: I have added it to the table.

- Paragraph 5 only discusses trastuzumab, but ends with a conclusion sentence on margetuximab. Maybe the title and the conclusion need to be changes, or the author needs to discuss the other "monoclonal antidoby-based therapies" too

Response: This paper aims to discuss margetuximab as a new therapy so adding more information about other monoclonal antibody-based therapy apart from what I wrote will make it a broader topic while I intend to direct the reader’s attention more to margetuximab. Accordingly, if you have any suggestions for the title, please let me know.

- The toxicity profile paragraph needs to be clearer: the data are presented in a misleading way and some additional comment from the trials data may be added

Response: Please check the Tolerability and toxicity paragraph in the revised manuscript.

- Table 2 needs to be updated, and data from the recently published updated OS of the SOPHIA trial need to be added

Response: I have added it to the table and the results have been discussed under section 6.6 lines (336-353).

- English needs to be revised for mistyping, grammar and syntax mistakes. 

Response: I have sought the help of a professional English language editing service to revise the manuscript for grammar and syntax errors as well as proofreading.

Round 2

Reviewer 2 Report

The author answered positively to all the questions raised by this reviewer, thus reaching a suitable form to be published.

Author Response

Thank you

Reviewer 3 Report

The updated version has solved most of my concerns. I would like for the author to address paragraph 5. I am not sure it's my place to suggest anything, but I would say the major options are either to add in the text what the paragraph title says, or to change the paragraph title to what it's actually written. Either way, or additional idea, the title and text of the paragraph need to match. 

Author Response

Thank you for your additional comments, I have changed the title to suit the main aim of writing this review.
